# CYP6AE gene cluster knockout in *Helicoverpa armigera* reveals role in detoxification of phytochemicals and insecticides

Huidong Wang[1], Yu Shi[1], Lu Wang[1], Shuai Liu[1], Shuwen Wu[1], Yihua Yang[1], René Feyereisen [2] & Yidong Wu [1]

The cotton bollworm *Helicoverpa armigera*, is one of the world's major pest of agriculture, feeding on over 300 hosts in 68 plant families. Resistance cases to most insecticide classes have been reported for this insect. Management of this pest in agroecosystems relies on a better understanding of how it copes with phytochemical or synthetic toxins. We have used genome editing to knock out a cluster of nine P450 genes and show that this significantly reduces the survival rate of the insect when exposed to two classes of host plant chemicals and two classes of insecticides. Functional expression of all members of this gene cluster identified the P450 enzymes capable of metabolism of these xenobiotics. The CRISPR-Cas9-based reverse genetics approach in conjunction with in vitro metabolism can rapidly identify the contributions of insect P450s in xenobiotic detoxification and serve to identify candidate genes for insecticide resistance.

[1] College of Plant Protection, Nanjing Agricultural University, 210095 Nanjing, China. [2] Department of Plant and Environmental Sciences, University of Copenhagen, Copenhagen 1017, Denmark. These authors contributed equally: Huidong Wang, Yu Shi. Correspondence and requests for materials should be addressed to Y.W. (email: wyd@njau.edu.cn)

Plants and insects have interacted on our planet for more than 400 million years. In the millions of years of what is often called an insect-plant arms race, plants have developed effective defense systems to protect them from insect herbivores with an array of detrimental plant secondary compounds, and herbivorous insects have accordingly evolved diverse mechanisms for detoxifying plant toxins[1]. Most herbivorous insect species are specialists feeding on a small number of related host plants or one or two plant families. The specialists reliably detoxify the chemical defenses that are common to their few hosts. A minority of herbivores are generalists, which can feed on a wider variety of plants and possess the ability to deal with a greater diversity of chemical defenses[2]. The specialists typically display rapid and efficient mechanisms for detoxification and/or sequestration of specific toxins of their phytochemically similar host plants. For instance, *Papilio polyxenes*, which specializes on Rutaceae and Apiaceae, encounters and tolerates concentrations of up to 1% of furanocoumarins in its diet with the aid of the detoxifying enzymes *CYP6B1* and *CYP6B3*[3–5]. But how generalist arthropods with hosts in multiple, chemically distinct plant families cope with phytochemical diversity at the molecular level is less clear[6].

Insect detoxification can be divided into three phases, phase I (oxidation, hydrolysis, reduction), phase II (conjugation) and phase III (excretion). The multifunctional activities of cytochrome P450 monooxygenases (P450s) in phase I reactions contribute to biochemical defense mechanisms against natural and synthetic toxins in insects, from host phytochemicals to synthetic insecticides[1,2,7–9]. Insect genomes can carry from lows of 36 CYP genes in the body louse[10] and 46 CYP genes in the honey bee[11] to highs of 163 CYP genes in the brown marmorated stink bug[12] and 170 CYP genes in the *Culex* mosquito[13]. Despite the various sizes of CYPomes in insects, many genes, often of the CYP 3 and CYP 4 clans, are arrayed in tight clusters of tandemly duplicated genes, reflecting recent duplications. The presence of multiple, closely related CYP genes in the genomes of pest insects presents a challenge to the functional identification of the genes that are important in adaptation to plant chemicals and detoxification of insecticides.

However, the recently developed CRISPR-Cas9 system has turned this challenge into an opportunity. The CRISPR-Cas9 system (clustered regularly interspaced short palindromic repeats/CRISPR-associated protein 9) is a powerful genome editing technology[14]. CRISPR-Cas9-directed gene insertions or deletions have been successfully employed to create genetic modification through non-homologous end joining (NHEJ) or homology-directed repair (HDR) after CRISPR-Cas9-directed double-stranded breaks (DSB). It has also been used to generate large-scale genomic deletions via the dual sgRNA system[15]. Therefore, the CRISPR-Cas9 system provides a promising tool for genetic manipulation of P450 clusters.

The cotton bollworm, *Helicoverpa armigera* (Hübner), as one of the world's most important agricultural pests, is distributed in Europe, Africa, Asia, and Australia, and has recently expanded in South America[16]. *H. armigera* is highly polyphagous and its host plants are recorded from 68 families worldwide[17]. Previous studies have indicated that several *CYP6AE* genes of *H. armigera* are inducible by plant toxins and insecticides, suggesting their potential role in detoxification[18,19]. *CYP6AE14* was suggested as a candidate P450 gene for detoxification of the cotton phytochemical gossypol in *H. armigera* as a result of experiments showing RNAi-mediated impaired larval tolerance to gossypol[20]. However, while in vitro metabolism studies showed that most of the CYP6AE family members can metabolize the pyrethroid insecticide esfenvalerate, none, including CYP6AE14, was able to metabolize gossypol[21,22]. The CYP6AE subfamily contains 10 P450 genes, nine of them are arranged as a large cluster on the chromosome 16 of *H. armigera*[22]. Thus, we felt that knockout of the CYP6AE cluster in vivo by CRISPR-Cas9 could provide reverse genetics evidence for the involvement of the CYP6AE cluster in xenobiotic detoxification in this species.

In this study, we present the results of this approach. Firstly, we deleted a 85 kb genomic fragment covering all nine P450 genes from the CYP6AE subfamily by the CRISPR-Cas9 method. We then assayed diverse phytochemicals and insecticides against both the background strain and the knockout strain. Secondly, we functionally expressed recombinant CYP6AE enzymes to identify the specific P450s involved in detoxifying xenobiotics. The present study not only identified several of CYP6AEs involved in detoxifying plant toxins and chemical insecticides, but also clarified that the CYP6AE cluster is not involved in gossypol defense in *H. armigera*.

## Results

**A strain homozygous for deletion of the CYP6AE cluster**. Dual sgRNA-directed CRISPR-Cas9 system was employed to delete the whole CYP6AE cluster. According to the genomic arrangement of CYP6AE cluster (Fig. 1a, b), the sgRNA14 and sgRNA12 were designed to target the genes at each end of the cluster, *CYP6AE14* and *CYP6AE12* respectively. The two specific sgRNAs and Cas9 protein were co-injected into *H. armigera* early embryos. Among the 400 injected eggs, 31% (125/400) hatched. Among the 125 neonates, 52% (65/125) developed into adults ($G_0$). The 20 female adults of $G_0$ were single pair mated with 20 male adults of the susceptible SCD strain and vice versa. Among the 40 single pairs, 18 single pairs produced fertile offspring ($G_1$). Ten second instar larvae from each of the 18 single pair families were pooled to prepare genomic DNA samples. PCR amplification with the primer pair 14 F/12 R revealed a ~600 bp band was present in the $G_1$ progeny from one of the 18 single pair families. Sequencing of the ~600 bp fragment confirmed a deletion event of CYP6AE cluster was created and inherited in that single pair family. Thus other larvae from this single pair family were reared to pupation, and 96 pupae were nondestructively genotyped for the CYP6AE cluster deletion with exuviates of the final instar larvae[23]. Thirty seven out of the 96 pupae genotyped were heterozygous for the deletion of CYP6AE cluster, and the 37 individuals (20 males, 17 females) were mass crossed to produce $G_2$. Ninety-six individuals of $G_2$ were genotyped, and 15 individuals (8 males and 7 females) homozygous for the CYP6AE cluster deletion were mass crossed to generate a homozygous knockout strain named as SCD-d6AE. The viability of this strain under our laboratory rearing conditions indicates that none of the *CYP6AE* genes is essential for *H. armigera* survival.

**Susceptibility to phytochemicals and insecticides**. Responses of larvae from the knockout strain and the background SCD strain to a series of concentrations of each phytochemical and insecticide were determined. We used the conservative criterion of non-overlap of 95% fiducial limits to assess differences in $LC_{50}$ for pairwise comparisons between strains.

Compared with the SCD strain, $LC_{50}$ values were significantly decreased to xanthotoxin (4.1-fold) and 2-tridecanone (1.4-fold) in the knockout SCD-d6AE strain (Fig. 2, Supplementary Table 1). The $LC_{50}$ values of three other phytochemicals (gossypol acetate, nicotine, coumarin) did not differ significantly between the two strains. It shows the CYP6AE cluster contributes to tolerance to xanthotoxin and 2-tridecanone, but not to gossypol acetate, nicotine, or coumarin.

$LC_{50}$ values of the knockout strain decreased significantly for esfenvalerate (4.5-fold) and indoxacarb (3.1-fold) when compared

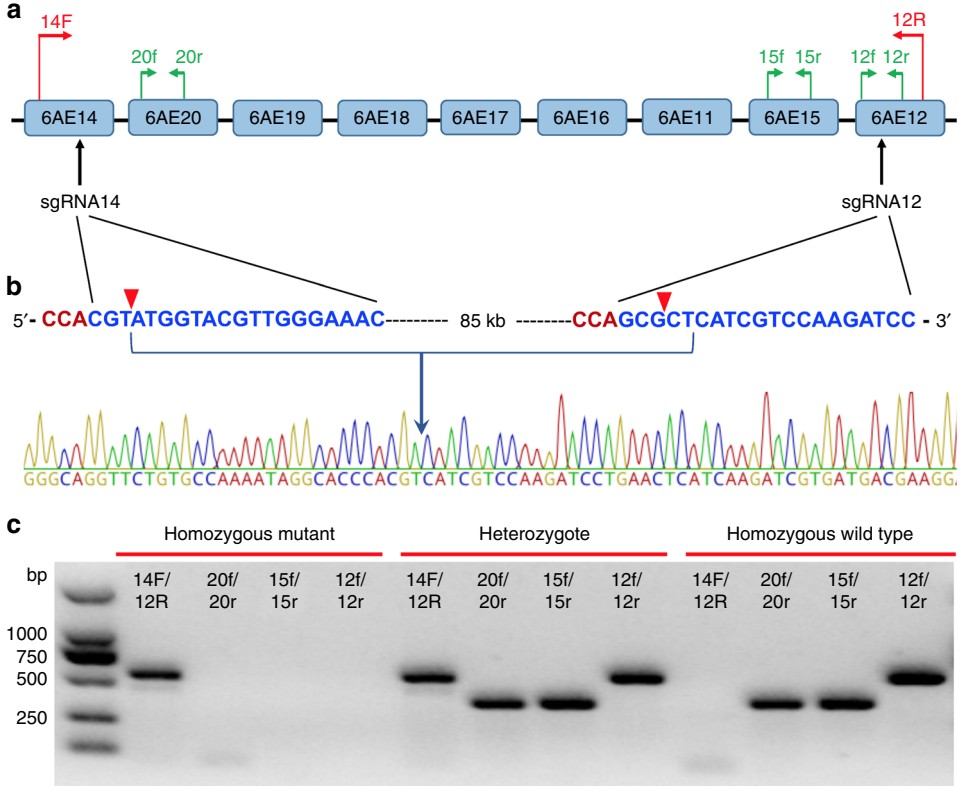

**Fig. 1** CRIPSPR-Cas9-based knock out of the CYP6AE cluster of *H. armigera*. **a** Positions of the two sgRNAs (sgRNA14 and sgRNA12) and the four primer pairs for allele-specific PCR detection. **b** Target sequences of the two sgRNAs (blue), the PAM sequences (red), and a representative chromatogram of direct sequencing of PCR products of individuals from the SCD-d6AE strain with the primer pair 14F/12R. The cutting sites by the Cas9 protein are indicated with red triangles. **c** Genotyping of individual *H. armigera* for deletion of the CYP6AE cluster according to banding patterns of the PCR products amplified with a set of four primer pairs

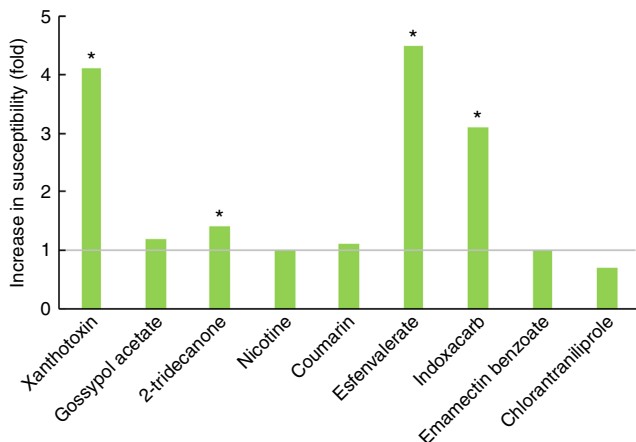

**Fig. 2** Responses to phytochemical toxins and insecticides (SCD-d6AE relative to SCD). The SCD-d6AE strain was derived from SCD by knocking out the CYP6AE cluster. Increased folds in susceptibility were calculated as $LC_{50}$ of SCD/$LC_{50}$ of SCD-d6AE. $LC_{50}$ values were considered significantly different if their fiducial limits did not overlap, and asterisks indicate statistical significance of $LC_{50}$s between the two strains. Phytochemical toxins: xanthotoxin, gossypol acetate, 2-tridecanone, nicotine, and coumarin. Insecticides: esfenvalerate, indoxacarb, emamectin benzoate, and chlorantraniliprole

with the SCD strain (Fig. 2, Supplementary Table 2). However, $LC_{50}$ values for emamectin benzoate or chlorantraniliprole did not differ significantly between the two strains. These results indicate that the CYP6AE cluster is involved in the tolerance to esfenvalerate and indoxacarb.

**Metabolism of chemicals by recombinant CYP6AE P450s.** Bioassay data of the SCD strain and the CYP6AE cluster knockout strain demonstrated that the CYP6AE cluster contributes to the tolerance of two phytochemicals (xanthotoxin and 2-tridecanone) and two insecticides (esfenvalerate and indoxacarb). Our previous study showed that all CYP6AE P450s except CYP6AE20 can metabolize esfenvalerate with different activities[22]. To identify which CYP6AEs contribute to detoxification, all ten P450 members of the CYP6AE subfamily were functionally expressed and in vitro metabolism of xanthotoxin, 2-tridecanone, and indoxacarb was investigated.

Ten *H. armigera* CYP6AE P450s and *Papilio polyxenes* CYP6B1 were recombinantly co-expressed with *H. armigera* cytochrome P450 reductase (HaCPR) in High Five cells using baculovirus expression system. Successful expression of CYP6AE P450s was demonstrated by their reduced CO-difference spectrum and model substrate activity tests[22]. While higher enzyme activities might be achieved with different experimental conditions (multiplicity of infection, addition of cytochrome b5), the in vitro assays allow comparisons of the catalytic competence

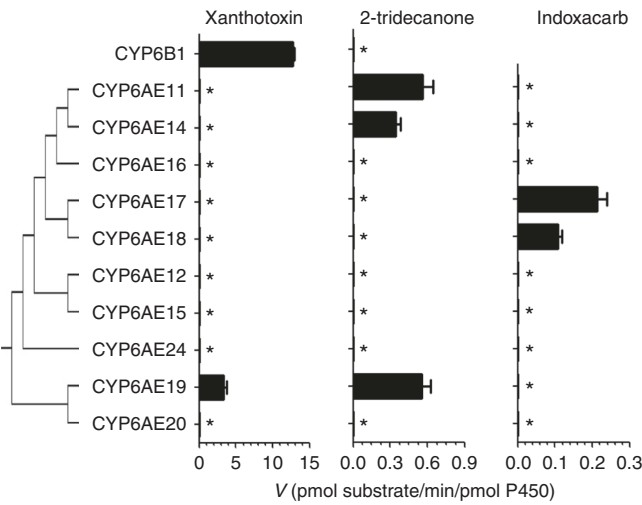

**Fig. 3** Metabolism of xanthotoxin, 2-tridecanone, and indoxacarb by recombinant P450s. CYP6AE subfamily enzymes were from *H. armigera* and CYP6B1 from *Papilio polyxenes*. Error bars represent mean values ± SEM ($n = 4$). Asterisk indicates that no significant metabolism was detected (limits of detection were 0.081, 0.033, and 0.016 pmol/min/pmol P450 for xanthotoxin, 2-tridecanone, and indoxacarb, respectively)

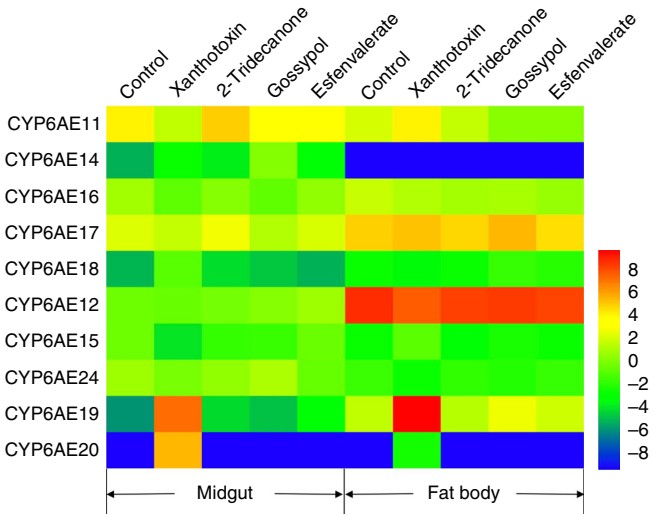

**Fig. 4** Effects of diverse xenobiotics on mRNA levels of CYP6AE subfamily genes. The midgut and fat body of the *H. armigera* were from the SCD strain of *H. armigera*. The mRNA levels were determined by RT-qPCR. Each gene has 5 biological repeats. The heat map was generated using the log2-transformed expression ratios relative to the midgut mRNA expression of CYP6AE12 in the control group. Blue: low expression levels. Red: high expression levels

of the individual CYP6AE P450s. CYP6B1 was also successfully expressed in High Five cells. The reduced CO-difference spectrum of CYP6B1 showed a distinct peak at 450 nm and indicated high quality recombinant P450 protein (~200 pmol/mg P450 in prepared microsomes).

As a positive control, *P. polyxenes* CYP6B1 showed the highest activity observed in the P450s we tested towards xanthotoxin, at 12.71 ± 0.23 pmol/min/pmol P450. Among the *H. armigera* CYP6AE P450s, only CYP6AE19 was shown to metabolize xanthotoxin with an activity of 3.32 ± 0.43 pmol/min/pmol P450. No significant decrease of xanthotoxin could be detected with any of the other CYP6AE P450s (Fig. 3 and Supplementary Figure 1, Supplementary Table 3).

2-tridecanone was metabolized by three CYP6AE P450s: CYP6AE11, CYP6AE14, and CYP6AE19, and their enzyme activities ranged from 0.34 to 0.56 pmol/min/pmol P450 (Fig. 3 and Supplementary Figure 2). CYP6B1, reported as a selective detoxification enzyme to xanthotoxin, showed no metabolic ability towards 2-tridecanone.

Indoxacarb was metabolized by CYP6AE17 and CYP6AE18 at 0.21 ± 0.03 pmol/min/pmol P450 and 0.11 ± 0.01 pmol/min/pmol P450 respectively (Fig. 3 and Supplementary Figure 3).

**Induction of expression of CYP6AE genes by xenobiotics.** Induction of expression of the CYP6AE subfamily P450 genes in response to dietary exposure of larvae to three allelochemicals (xanthotoxin, gossypol, 2-tridecanone) and an insecticide (esfenvalerate) was measured in the SCD strain with RT-qPCR.

The results (Fig. 4, Supplementary Table 4) showed that the *CYP6AE19* and *CYP6AE20* transcripts (>1000-fold induction in midgut and >100-fold induction in fat body) are very strongly induced by xanthotoxin. Furthermore, *CYP6AE14* transcripts are induced in midgut (40-fold) after dietary exposure to gossypol.

The above metabolism study identified *CYP6AE19* as the sole P450 in the CYP6AE subfamily which is capable of detoxifying xanthotoxin. High inducibility of *CYP6AE19* by xanthotoxin demonstrates that *CYP6AE19* can be expressed at high levels and can metabolize xanthotoxin efficiently when needed.

## Discussion

The CRISPR-Cas9 system has been successfully used to edit target genes in many insect species including *H. armigera*[23]. There are 114 P450 genes in *H. armigera*[24], making genome editing of each gene a difficult task. However, we have taken advantage of the common clustering of these genes and in the present study, we successfully applied this genome editing system to produce a large deletion (85 kb) of the entire CYP6AE cluster of nine genes in *H. armigera*. We recovered a homozygous line with the large deletion from one of the 18 single-pair families made between moths of G0 and the background SCD strain, suggesting that at least 2.8% (1/36) of the CYP6AE cluster deletion allele was induced and transmitted to the next generation with the CRISPR-Cas9 system in the current study. This demonstrates that this system can be useful in P450 cluster knockout for *H. armigera* and for other insects in which the CRISPR-Cas9 system is effective. Knockout of the CYP6AE cluster does not affect viability of the insect under our rearing conditions, but it results in increased susceptibility to both plant toxins and synthetic insecticides. The knockout strain can now be used to compare fitness of this polyphagous insect on host plants that differ in their secondary chemistry. Differences would indicate the role of the CYP6AE cluster in tolerance to such plants. We complemented our in vivo data with in vitro metabolism using heterologous expression of each of the P450 enzymes of the cluster. We thus identified one or more specific CYP6AEs that can metabolize xanthotoxin, 2-tridecanone or indoxacarb. Previous studies showing induction of some *CYP6AE* genes by specific chemicals or different host plants[18,19,24], their constitutive overexpression in resistant strains[25], or their ability to metabolize xenobiotics in vitro[22] all suggested a role of the CYP6AE subfamily in detoxification. The CRISPR-Cas9-based reverse genetics approach has now provided in vivo data to strongly confirm that the CYP6AE subfamily cluster is involved in xenobiotic tolerance in *H. armigera*. Future work can aim at knocking out individual P450 genes of a cluster, and thus fine-tune the analysis of host plant tolerance phenotypes of each gene. However, our study is both a proof of principle as

well as a rapid screen taking advantage of the clustering of P450 genes.

A striking characteristic of CYPomes is the presence of one or more subfamilies that appear both abundant and lineage-specific, and that have been called CYP blooms[26]. In *H. armigera*, there are 11 subfamilies possessing three or more genes, of which nine form tight head to tail clusters[24]. These clusters belong to the CYP Clans 3 and 4, which comprise many genes related to xenobiotic detoxification although the clustered CYP4G genes are related to cuticular hydrocarbon biosynthesis. Many P450 clusters could be studied with the CRISPR-Cas9 mediated cluster deletion approach we developed in this study. Previous studies of insect species with reduced CYPome sizes such as the honey bee[11] and the human body louse[10], noted that functional studies would be facilitated by smaller and manageable numbers of CYP genes. This would make such species efficient models for xenobiotic metabolism, and rational insecticide design. Our research strategy shows that knocking out multiple detoxification genes with genome editing, even in species with large CYPomes, can serve as a first pass screen (or multiplexing) for tolerance to xenobiotics. A detailed analysis of the cluster by knocking out and functionally expressing individual genes can then follow.

Xanthotoxin is a linear furanocoumarin with high toxicity to a wide variety of organisms, including bacteria, nematodes, insects, fish, and mammals, because furanocoumarins can cause mutations in DNA and interfere with transcription after photo-activation. In non-adapted organisms they can also cause mechanism-based inhibition of P450 enzymes[27,28]. Insect herbivores show varying degrees of tolerance to furanocoumarin-containing plants. Adapted specialists such as *Papilio polyxenes* and less adapted generalists of the genus *Papilio* all rely on induction of, and metabolism by, P450 enzymes of the CYP6B subfamily[3–5,29–31]. However, induction by furanocoumarins and their metabolism appears more widespread in insects and involves CYP6AB and CYP321A subfamilies as well[32–34]. We now add *CYP6AE19* of *H. armigera* to the list of insect P450s capable of metabolizing xanthotoxin.

The *CYP6AE19* transcripts are expressed constitutively at low levels, but can be induced to extremely high levels (more than 1000-fold in midgut and 100-fold in fat body) after feeding with sublethal dose of xanthotoxin. As furanocoumarins are mechanism-based inhibitors of P450 enzymes, our heterologous expression system can now be used to compare the relative sensitivity to inhibition of *CYP6B1*, from a well adapted insect and *CYP6AE19* from a less adapted insect. Perhaps the very high induction levels of *CYP6AE19* we have seen are a way to compensate for higher inactivation of the P450 by host plant furanocoumarins.

We also found that *CYP6AE20* transcripts were strongly induced by xanthotoxin although to a lower abundance than *CYP6AE19*. However, we did not detect xanthotoxin metabolism by *CYP6AE20*. *CYP6AE19/20* are the closest members of the cluster, being 91% identical at the protein level[22], and they may have similar promoter region allowing for induction, yet a different biochemical function. This inconsistency between the metabolism capability and inducibility confirms that inducibility is not necessarily correlated to detoxification capability. Therefore inducibility alone cannot be used as evidence for a role in detoxification and a comprehensive analysis such as reverse genetics and/or functional expression will be needed when considering xenobiotics detoxification by herbivore P450s.

Mao et al.[20] reported that silencing *CYP6AE14* gene by plant-mediated RNAi impairs larval tolerance of gossypol, suggesting the *CYP6AE14* could detoxify gossypol. But later studies showed that *CYP6AE14* and in fact none of the CYP6AE subfamily enzymes could metabolize gossypol in heterologous expression

systems[21,22]. To clarify these contrasting conclusions, we knocked out the CYP6AE gene cluster in vivo and tested whether gossypol susceptibility was affected in the knockout strain. However, no susceptibility change was observed after knockout of the CYP6AE cluster, which is consistent to previous in vitro metabolism[22]. The possible reasons of the discrepancy with the results of Mao et al.[20] could include off-target effects, unknown fitness costs, sequestration[22], or other detoxification mechanisms as revealed by Krempl et al.[35]. Our investigations by CRISPR-Cas9-mediated knockout of the CYP6AE cluster excluded the off-target effects of RNAi, because the possible off-target CYP6AE genes were knocked out simultaneously. Our results also probably exclude sequestration by CYP6AE14 or other CYP6AE enzymes, unless the effect of sequestration is only significant when protein levels are highly induced. Yet CYP6AE14 transcript levels are normally low, so even after induction its protein levels are not expected to be very high, certainly not as high as the protein levels of sequestering esterases in mosquitoes for instance.

In this study, susceptibility to esfenvalerate was significantly increased (4.5-fold at $LC_{50}$) in the CYP6AE cluster knockout strain, which confirmed the previous results from in vitro metabolism[22]. We also found that susceptibility to another insecticide, indoxacarb, was significantly increased after the CYP6AE cluster was knocked out, suggesting that one or more CYP6AE enzymes from the cluster can metabolize indoxacarb. Previous studies have suggested that indoxacarb resistance is associated with enhanced metabolism by P450 monooxygenases in *Choristoneura rosaceana*[36], *Spodoptera litura*[37] and *H. armigera*[38]. In the GY7–39 strain of *H. armigera*, indoxacarb resistance was autosomal, incompletely dominant, and the resistance was suppressed by the P450 inhibitor PBO[38]. Our results for indoxacarb metabolism could offer a valuable clue for investigating indoxacarb resistance mechanisms in the GY7–39 strain, and it would be worthwhile comparing expression levels of *CYP6AE17* and *CYP6AE18* between indoxacarb-resistant and susceptible strains of *H. armigera*. While our approach provides candidate genes for insecticide resistance (an a priori approach), it is clear that P450 genes other than the ones we knocked out, indeed non P450 genes may also be involved in field-evolved resistance. It is also possible that CYP6AE genes would be involved in field resistance to indoxacarb in some populations and not in others. Reverse genetics can also be used a posteriori to confirm the role of a P450 in resistance as for *CYP9M10* involved in pyrethroid resistance in a *Culex* mosquito[39].

In the discussion of plant-herbivore interactions, association with a particular chemically distinct group of host plant taxa is considered an evolutionary innovation. Specialization on furanocoumarin-containing plants and conservation of P450 detoxification of furanocoumarins has been identified as a key innovation within the genus *Papilio*[40]. Wheat et al.[41] identified a pierid glucosinolate detoxification mechanism, nitrile-specifier protein (NSP) as a key innovation in a butterfly-host plant system. We propose that clustering of detoxification enzymes may be seen as key innovation in the exploration of chemical space by herbivores. Since their discovery[42] clusters of P450 genes have become a landmark of insect CYPomes. In Lepidoptera, clusters such as the CYP6AE cluster are found in both butterflies and moths, and may have originated over 150 million years ago. The fact that these clusters are maintained despite very high rates of genomic rearrangements seen in Lepidoptera[43] would indicate selection against the dispersion of their genes throughout the genome. Furthermore, it is the clustering itself which is maintained, rather than the identity of the genes that compose them. For instance, of nine CYP9A genes in a *Spodoptera friguperda* cluster, only four can be considered orthologous to CYP9A genes of *H. armigera*[26]. Clustering of CYP2 clan genes such as CYP306

and CYP18, involved in ecdysone synthesis and catabolism respectively, might be explained as a way to favor inheritance of the two genes together, thus avoiding deleterious hormonal imbalance. For the CYP6AE cluster, complete deletion does not entail an obvious fitness deficit, and consumption of different host plants would select for different component genes because our results with those of Shi et al.[22] show that the nine P450 of this cluster each have a specific catalytic competence towards 10 different substrates. We have shown here that while CYP6AE19 can metabolize both xanthotoxin and 2-tridecanone, the latter is also metabolized by CYP6AE11 and CYP6AE14. *H. armigera* can be found as pest on citrus and tomato and thus encounters non simultaneously a wide variety of potentially toxic phytochemicals including furanocoumarins such as xanthotoxin in citrus (Rutaceae) and 2-tridecanone in tomato (Solanaceae). Selection of the cluster as a heritable unit would maintain this pests' ability to switch host plant, at least between Rutaceae and Solanaceae. In this broadly adaptationist view, selection would act less on a specific pair of toxin / detoxifying enzyme than on the maintenance of catalytic promiscuity of the enzymes encoded by clustered genes. This promiscuity, in turn, can be exploited when facing new chemical challenges such as pesticides.

The reverse genetics approach presented here will allow a critical evaluation of this view of CYP cluster evolution. When combined with functional expression of P450 enzymes, it will also accelerate the process of identifying key players in insecticide metabolism which can be considered candidate genes for resistance.

## Methods

**Insect strains and rearing.** The wild-type strain SCD that was originally collected from Côte D'Ivoire (Ivory Coast, Africa) in the 1970s was kindly provided by Bayer Crop Science in 2001[44]. This strain has been maintained in the laboratory without exposure to insecticides for more than 40 years. The SCD-d6AE strain was derived from SCD by knocking out the whole CYP6AE cluster with the CRISPR-Cas9 genome editing tool (see below). This knockout strain is a deficiency strain for a cluster of nine P450 genes (specifically *CYP6AE14, CYP6AE20, CYP6AE19, CYP6AE18, CYP6AE17, CYP6AE16, CYP6AE11, CYP6AE15,* and *CYP6AE12*).

Larvae of the two strains were reared on an artificial diet based on wheat germ and soybean powder under conditions of $26 \pm 1 °C$, $60 \pm 10\%$ relative humidity and a photoperiod of 16 h light: 8 h dark. 10% sugar solution was supplied for adults.

**Plant allelochemicals and synthetic insecticides.** For bioassays: xanthotoxin (98%, J&K Chemical Ltd., Shanghai, China), gossypol acetate (98%, Ciyuan Biotechnology Co. Ltd., Shaanxi, China), 2-tridecanone (99%, Sigma-Aldrich company, Saint Louis, MO), coumarin (99%, Sigma-Aldrich company, Saint Louis, MO), nicotine (99.9%, Duly Biotech Co. Ltd, Jiangsu, China), esfenvalerate (99%, Aladdin Industrial Corporation, shanghai, China). Insecticide formulations of indoxacarb (50 g/l EC), emamectin benzoate (20 g/l EC), and chlorantraniliprole (50 g/l EC) were provided by the Plant Protection Institute, Guangdong Academy of Agricultural Sciences, Guangzhou, China.

For in vitro metabolism: xanthotoxin (99.98%, MedChemExpress company, Monmouth Junction, New Jersey), 2-tridecanone (99%, Sigma-Aldrich company, Saint Louis, MO), and indoxacarb (R/S = 1:1.05, 99%, Dikma Technologies, Foothill Ranch, CA). HPLC solvents were purchased from Fisher Scientific (Pittsburgh, PA).

**Preparation of single guide RNA (sgRNA).** A PCR-based approach was used to prepare sgRNA according to the manufacturer's instruction (GeneArt™ Precision gRNA Synthesis Kit, Thermo Fisher Scientific, Pittsburgh, PA). Briefly, a forward oligonucleotide (TAATACGACTCACTATAG + target sequence) and a reverse oligonucleotide (TTCTAGCTCTAAAAC + target sequence reverse complement) were assembled by PCR with the Tracr Fragment + T7 Primer Mix to generate sgRNA DNA template. The PCR assembly reaction mixture (25 µl) contained 12.5 µl of Pfusion™ High-Fidelity PCR Master Mix, 1 µl of Tracr Fragment + T7 Primer Mix, 1 µl of 0.3 µM Target F1/R1 oligonucleotide mix, 10.5 µl Nuclease-free water. PCR was performed at 98 °C 10 s, 32 cycles of (98 °C 5 s, 55 °C 15 s), 72 °C 1 min and 4 °C∞. Then in vitro transcription (IVT) reaction was performed to generate sgRNA. The IVT reaction mixture (25 µl) contained 8 µl NTP mix, 6 µl gRNA DNA template, 4 µl 5X TranscriptAid™ Reaction Buffer, 2 µl TranscriptAid™ Enzyme Mix. After generating sgRNA by IVT, the DNA template was removed by DNase I digestion, and sgRNA was purified by the gRNA Clean Up Kit. Primer sequences used were detailed in Supplementary Table 5.

**Embryo microinjection.** The collection and preparation of eggs were carried out as reported by Wang et al.[23] Briefly, fresh eggs laid within 2 h were washed down from the gauzes in 1% sodium hypochlorite solution and rinsed with distilled water. After suction filtration, the eggs were lined up on a microscope slide (fixed with double-sided adhesive tape).

About one nanoliter mix of sgRNA14 (100 ng/µl), sgRNA12 (100 ng/µl) and Cas9 protein (100 ng/µl, GeneArt™ Platinum™ Cas9 Nuclease, Thermo Fisher Scientific, Shanghai, China) were injected into individual eggs using a FemtoJet and InjectMan NI 2 microinjection system (Eppendorf, Hamburg, Germany). The microinjection was completed within 2 h. Injected eggs were placed at 26 ± 1 °C, 60 ± 10% RH for hatching.

**Identification of deletion mutation of the CYP6AE cluster.** According to the *CYP6AE14* and *CYP6AE12* gene orientations in chromosome 16 and sgRNA locations within each gene, the forward primer 14 F (located in *CYP6AE14*) and the reverse primer 12 R (located in *CYP6AE12* gene) were designed to detect the CYP6AE cluster deletion (Fig. 1a, b). A small fragment of genomic DNA (about 600 bp) is expected to be amplified with the primer pair 14F/12R if the CYP6AE cluster is deleted. To determine whether the CYP6AE cluster deletion mutation is homozygous or heterozygous, three pairs of specific primers were designed for the three P450 genes within the CYP6AE cluster. The primer pair *20 f/r* is specific to *CYP6AE20* gene (a PCR product of 396 bp), the primer pair *15 f/r* is specific to *CYP6AE15* gene (391 bp), and the primer pair *12 f/r* is specific to *CYP6AE12* gene (582 bp). Genotypes of the CYP6AE cluster deletion mutation can be discriminated according to the banding pattern of PCR amplified products (Fig. 1c).

**Bioassays.** Toxicity of each phytochemical to the SCD and SCD-d6AE strains was determined with diet incorporation bioassays. Gradient concentrations of each testing phytochemical were mixed thoroughly in liquid artificial diet. Liquid artificial diet (1 ml) was dispensed into each well of a 24-well plate. After the diet cooled and solidified, a single unfed neonate (24 h old) was put in each well, and 24–48 larvae of each strain were tested for each concentration. Five to seven concentrations of each chemical were used to establish the log-probit lines. Mortality was recorded after 5 days. Larvae were scored as dead if they died or did not reach the second instar at the end of bioassays.

Toxicity of each insecticide to the SCD and SCD-d6AE strains were determined with diet surface overlay bioassays. The formulated insecticides were diluted to generate serial dilutions with distilled water. Liquid artificial diet (1 ml) was dispensed into each well of a 24-well plate. After the diet cooled and solidified, 100 µl of insecticide solution was applied evenly to the diet surface in each well and allowed to dry at room temperature. One second-instar larva was placed in each well of a 24-well plate, and 48 larvae of each strain were tested at each concentration. Five to eight concentrations of each insecticide were used to establish the log-probit lines. Mortality was recorded after 2 days for esfenvalerate, indoxacarb and chlorantraniliprole, and 3 days for emamectin benzoate. Larvae were scored as dead if they did not respond after gentle prodding.

All tests were done at $26 \pm 1 °C$, $60 \pm 10\%$ relative humidity and a photoperiod of 16 h light: 8 h dark except xanthotoxin was tested under dark conditions. The $LC_{50}$ values (the concentration killing 50% of larvae) and the 95% fiducial limits of the $LC_{50}$ for each strain were calculated through probit analysis of the mortality data using the PoloPlus program[45]. Two $LC_{50}$ values were considered significantly different if their 95% fiducial limits did not overlap[46].

**In vitro metabolism of chemicals by CYP6AE enzymes.** The open reading frames (ORFs) of 10 P450 genes from the CYP6AE subfamily were cloned from SCD strain and expressed in High Five cells by Bac-to-Bac system (Invitrogen)[22]. *Papilio polyxenes CYP6B1* (*CYP6B1v1*, GenBank accession no. M80828) was obtained by gene synthesis. All P450s were co-expressed with HaCPR with multiplicity of infection (MOI) of 2 and 0.2 respectively. The content of recombinant P450s in microsomal protein were determined by reduced CO-difference spectra assay[47].

In vitro metabolism was performed with 20 pmol recombinant P450, NADPH regeneration system (Promega, Madison, WI) and 2 µl substrate (20 µM xanthotoxin dissolved in DMSO, 20 µM 2-tridecanone or 10 µM indoxacarb dissolved in acetonitrile) in 200 µl 0.1 M potassium phosphate buffer (pH 7.4). Reactions were pre-warmed in a 30 °C heating block for 5 min and started after adding substrate. Samples were incubated on an orbital shaking incubator at 30 °C, 1200 rpm (30 min for xanthotoxin metabolism, 1 h for 2-tridecanone and indoxacarb metabolism). Reactions were stopped by adding 200 µl acetonitrile, then 600 µl dilution buffer (50% potassium phosphate buffer and 50% acetonitrile) was added to each sample and incubated for further 20 min. The stopped reactions were centrifuged at $18,000 \times g$ for 10 min, 800 µl cleaned supernatant was transferred to HPLC vial and analyzed immediately. Negative control with equivalent total protein of non-insertion microsomes were tested at the same time for each P450. CYP6B1 was used as positive control in the metabolism of xanthotoxin. Samples without NADPH were performed at the same time. Experiments were repeated twice with different batches of recombinant P450s.

**UPLC-MS/MS**. The clearance of substrates was applied as the criterion of the enzyme activity. Samples were detected by UPLC-MS/MS (Waters Xevo TQ-S micro, MA) with MRM and run in positive ESI mode (parameters were showed in Supplementary Table 6). Samples were separated by BEH C8 column (2.1 × 50 mm, 1.7 μm particle size, Waters, MA) on UPLC system (Waters ACQUITY UPLC I-Class) with different gradient elution programs, the flow speed was 0.3 ml/min.

For metabolism of xanthotoxin, 1 μl samples were separated with acetonitrile/0.1% (v/v) formic acid and water/0.1% (v/v) formic acid. The gradient elution program was: 0 min acetonitrile:H₂O 10:90, 0.3 min acetonitrile:H₂O 10:90, 2 min acetonitrile:H₂O 95:5, 3 min acetonitrile:H₂O 95:5, 3.1 min acetonitrile:H₂O 10:90, 5 min acetonitrile:H₂O 10:90.

For metabolism of 2-tridecanone, 1 μl samples were separated with acetonitrile and water/0.1% (v/v) formic acid. The gradient elution program was: 0 min acetonitrile:H₂O 20:80, 0.3 min acetonitrile:H₂O 20:80, 1 min acetonitrile:H₂O 90:10, 3 min acetonitrile:H₂O 100:0, 4 min acetonitrile:H₂O 100:0, 4.2 min acetonitrile:H₂O 20:80, 5.8 min acetonitrile:H₂O 20:80.

For metabolism of indoxacarb, 1 μl samples were separated with acetonitrile and water/0.1% (v/v) formic acid. The gradient elution program was: 0 min acetonitrile:H₂O 10:90, 0.3 min acetonitrile:H₂O 10:90, 1.5 min acetonitrile:H₂O 95:5, 2.5 min acetonitrile:H₂O 95:5, 2.6 min acetonitrile:H₂O 100:0, 3 min acetonitrile:H₂O 100:0, 3.1 min acetonitrile:H₂O 10:90, 5 min acetonitrile:H₂O 10:90.

Xanthotoxin, 2-tridecanone, and indoxacarb eluted at 1.77 min, 2.15 min, and 2.09 min respectively (Supplementary Figures 1, 2, 3).

The degradation of substrates was quantified with an external standard method. The linear calibration was used to calculate the limit of detection of each compound as three times the root mean square error of the y residuals divided by the slope of the regression line. Recovery rates of the three substrates using recombinant microsomal protein without NADPH regeneration system were close to 100%. The final metabolic activity was corrected by subtracting the background (potential decrease in negative control) and homogenized by pmol P450, and expressed as pmol substrate per minute per pmol P450.

**Induction treatment and RT-qPCR**. Induction treatment experiments were performed referring to the method from Zhou et al.[18]. Three plant allelochemicals (gossypol, xanthotoxin and 2-tridecanone) and an insecticide (esfenvalerate) were used as inducer in this study. Newly molted 5th instar larvae were fed with artificial diet containing different chemicals (1 mg/g for xanthotoxin, gossypol and 2-tridecanone respectively, and 5 μg/g for esfenvalerate). All chemicals were used at sublethal concentrations[18]. The chemicals were dissolved in acetone to make stock solutions. The stock solution was mixed with artificial diet at 10 μl per gram diet, and the corresponding volume of acetone was served as a control. After feeding for 48 h, both midgut and fat body were dissected for RNA extraction. Tissues from 10 individuals were pooled as one sample and 5 biological replicates were performed for all treatments.

Total RNA was extracted using TRIzol (Invitrogen, CA) according to the manufacturer's instructions. The cDNA was synthesized by using PrimeScript Reverse Transcriptase kit (TaKaRa, Shiga, Japan). Specific primers, reaction system and thermo-cycling program for each *CYP6AE* gene were the same used by Shi et al.[22]. The $2^{-\Delta\Delta CT}$ method[48] was used to calculate gene expression levels with the Ct values determined from RT-qPCR experiments. Data were normalized to the geometric mean of two housekeeping genes (actin and *EF-1α*).

**Reporting Summary**. Further information on research design is available in the Nature Research Reporting Summary linked to this article.

## Data availability
The authors declare that the data supporting the findings of this study are available within the paper and its Supplementary Information.

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

## Acknowledgements

This work was supported by grants from the Ministry of Agriculture of China (No. 2016YFD0200500), the Fundamental Research Funds for the Central Universities of China (KYT201803), and the Innovation Team Program for Jiangsu universities (No. 2013-6).

## Author contributions

Y.W., Y.Y. and S.W. designed research; H.W., Y.S., L.W. and S.L. performed research; Y.W., H.W., R.F. and Y.S. analyzed data; and H.W., Y.S., Y.W. and R.F. wrote the paper.

## Additional information

**Competing interests:** The authors declare no competing interests.

