## [Peer Review File · Nature Communications]

Reviewers' comments:

Reviewer #1 (Remarks to the Author):

The manuscript describes the functional analysis of a cluster of P450 genes in an insecticide sensitive strain of the cotton bollworm, *H. armigera* through genome editing, recombinant protein activity assays and transcriptomic analysis. Overall the major claim of the paper is that gene editing technology in non-model organisms now allows the characterisation of the global function of gene clusters through reverse genetic approaches. And when combined with proteomic and transcriptomic analysis can provide very strong evidence for the role of particular genes in the phenotypes assayed, in this case xenobiotic detoxification.

The P450 gene family is a prime target for such analysis, since particularly in insects, there has been major evolutionary pressure to create clusters (or blooms) of P450 genes that may be quite divergent in sequence and thus substrate specificity. The authors suggest that the maintenance of clusters implies a selective pressure to keep the substrate promiscuity of the inherited unit intact to allow adaption to varying food sources, that could also be exploited when exposed to different insecticides. As such the paper will be of particular interest to the large insecticide resistance community that encompasses major agricultural pests and public health disease vectors in which the different technologies utilised are either routine or are rapidly becoming established, and could be deployed in similar fashion to other arthropods of interest.

The particular novel aspect of the paper is the deletion of the entire cluster of related CYP6AE genes. Previous work had mainly used RNAi technology, and to a limited degree gene editing nucleases, to examine knockdown phenotypes of individual P450 genes in *H. armigera* and other non-drosophilid insects. By taking a more broader approach and deleting the whole cluster a number of advantages are apparent; these include

- i) allowing an initial global analysis of function of the entire gene cluster
- ii) establishing that some of the most important xenobiotic metabolising enzymes in the cluster are not vital to the organism, and if any of the genes perform critical housekeeping or intermediary metabolism then redundant function must lie in genes beyond this cluster.
- iii) diminishing the problem of individual gene knockdown or knockout in which closely related genes are upregulated in compensation that may mask or dilute phenotypes.

What the approach lacks is the ability to examine the role of individual genes, and this is clearly the next step in the research that would go on to use CRISPR/Cas9 to delete individual genes to reinforce the candidate gene phenotypes identified in the manuscript.

The paper's other novelty has been to challenge some previous conclusions identified through phenotypes produced by RNAi of individual genes, that were not supported by later enzyme activity data. The data also indicates that upregulation of specific P450s in response to xenobiotic exposure is not a guarantee that that P450 is involved in metabolism of the particular xenobiotic and may reflect co-regulation of genes within the cluster.

Before publication, I would like to see some modifications to the manuscript that would make the presentation a little clearer to those not intimate with the field. The level of detail provided in the manuscript is generally sufficient to reproduce the work, particularly if, as suggested below, that the indicated statistical methods were clarified.

These include:

- 1) I don't understand the prominence of the MS figures Fig4, 5 and 6. They are barely mentioned in the results and discussion and seem to be more suited to the supplementary material. I would

thus suggest that they were moved to supplementary or if there is further MS information that this space was used to identify the different xenobiotic metabolites to demonstrate compound detoxification.

2) Fig2 may be simplified to remove the reference strain values that by definition are equal to 1 in each case.

3) Supple table 1 should also present the raw data from the larval toxicity assays rather than just the summary data. Include in the table the precise statistical method used to produce the 95% fiducial limits for transparency and reproducibility by others.

4) An indication as to whether single gene knockouts will be performed or are necessary to validate results further.

Minor modifications:

Line 39 replace "responsible for detoxification" with capable of metabolism

135 replace detoxification of with tolerance to – the bioassays cannot show a detoxification route specifically

Ditto line 140 and 144 line

150 include *P. polyxenes* in front of CYP6B1

157 remove subjective values such as high – eg the value for *cyp6B1* is the highest observed in the samples tested.

line 160 put in the value of the next highest one to AE19 such that, only CYP6AE19 was shown to metabolise xanthotoxin at an activity >0.7 Pmol/

Line 162-165 These activity values indicate very slow turnover even for P450s. How significant are these values compared to the values given by asterix (which I presume are not zero?) Please indicate the cut off where activity is deemed to be negligible.

Line 190 Is there any evidence that viability of the knockout would be affected if fed certain foodstuffs. Possibly need to qualify sentence to under these experimental conditions.

Line 197 I think the following is correct; the reverse genetic approach has provided in vivo data to confirm involvement of the cluster in xenobiotic tolerance in *H. armigera*

Lines 212 215 I would argue that the strategy is not a precise way it provides a global view. The precise way would be to also knock out individual genes to give "full characterisation".

The global strategy is very useful as a first pass, along with transcriptomic and proteomic data.

Fig 7 indicate what the scale is on the right

Reviewer #2 (Remarks to the Author):

The authors used genome editing to knock out a cluster of nine cytochrome P450s (85kb) to demonstrate their cluster contribution to xenobiotic (phytotoxins and pesticides) detoxification, in a major agricultural pest world wide, *Helicoverpa armigera*. Validation was done by parallel in vitro functional studies (expression of recombinant enzymes and characterization of their catalytic properties against same xenobiotics used in bioassays), as well as induction and qPCR experiments.

Among claims, an interesting verification that the CYP6AE14 is not involved in gossypol defense (article previously published in Nature, possible RNAi leaking/off target effects). Another

interesting claim, the inducibility alone can not be used as evidence for a role in detoxification.

The findings are certainly novel and both outcome and methodology will be of substantial interest to others in the community and the wider field. The paper is going to really influence thinking in the field, and beyond (for example P450s involved in resistance of major vector borne diseases) as several classical approaches, linking gene to phenotype, will have to be re-considered.

The study is technically well executed and of very high quality standards and the manuscript is very well written and clearly presented. The conclusions are well supported by the experimental data and no further experiments are required. I would strongly recommend the acceptance of this manuscript in Nature Communication.

One minor thing would only recommend to the authors to include in their discussion, why they think there is still significant difference in the pesticide resistance levels observed in field populations in association studies (for example indoxacarb resistance and P450s), with the ones obtained after the KO (for example 3.1 – fold for indoxacarb).

Reviewer #3 (Remarks to the Author):

The article describes the use of CRISPR to delete a large cluster of P450 genes, that probably originally arose via duplication, with a view to investigating the role of members of this cluster in the detoxification of various insecticides and phytotoxins. The intro to the article and the description of the experimental flow are largely a pleasure to read for their flow and their clarity, so compliments to the authors for that. Having said that, functional confirmation of the role of P450s in such processes is not new, and neither is the use of CRISPR with two sgRNAs to perform large genomic deletions. Therefore I would have some significant concerns over the novelty of the findings.

What would have made this paper much stronger would be to have performed a functional evaluation of individual P450 genes *IN VIVO* in order to unequivocally tease out the roles of different P450s and to determine whether there has been specialisation of function. Instead, having performed the large deletion, the authors chose to tests different P450-toxin combinations *in vitro* and extrapolate their findings. The conclusions of these findings then made no use at all of the knockout and thus are subject to all the caveats of *in vitro* experiments such as controlling for copy number, tissue variegation, position effects at insertion site etc.

MINOR POINTS

Line 175 "based on Ct value" it is not clear from this statement how the quantification of the qRT experiments done and this is a peculiar choice of phrase.

Discussion is way too long. Lines 20-204 contain too much technical detail about a hypothetical future application of CRISPR editing. Lines 216-243 read more like a review article of P450s and are out of place here.

More might have been made about the frequency with which large deletions could be recovered and what this means for the field in general.

I am happy for my review to be non-anonymised.

Tony Nolan

Responses to comments from the referees

Reviewers' comments:

Reviewer #1 (Remarks to the Author):

The manuscript describes the functional analysis of a cluster of P450 genes in an insecticide sensitive strain of the cotton bollworm, *H. armigera* through genome

editing, recombinant protein activity assays and transcriptomic analysis. Overall the major claim of the paper is that gene editing technology in non-model organisms now allows the characterisation of the global function of gene clusters through reverse genetic approaches. And when combined with proteomic and transcriptomic analysis can provide very strong evidence for the role of particular genes in the phenotypes assayed, in this case xenobiotic detoxification.

The P450 gene family is a prime target for such analysis, since particularly in insects, there has been major evolutionary pressure to create clusters (or blooms) of P450 genes that may be quite divergent in sequence and thus substrate specificity. The authors suggest that the maintenance of clusters implies a selective pressure to keep the substrate promiscuity of the inherited unit intact to allow adaptation to varying food sources, that could also be exploited when exposed to different insecticides. As such the paper will be of particular interest to the large insecticide resistance community that encompasses major agricultural pests and public health disease vectors in which the different technologies utilised are either routine or are rapidly becoming established, and could be deployed in similar fashion to other arthropods of interest.

The particular novel aspect of the paper is the deletion of the entire cluster of related CYP6AE genes. Previous work had mainly used RNAi technology, and to a limited degree gene editing nucleases, to examine knockdown phenotypes of individual P450 genes in *H. armigera* and other non-drosophilid insects. By taking a more broader approach and deleting the whole cluster a number of advantages are apparent; these include

- i) allowing an initial global analysis of function of the entire gene cluster
- ii) establishing that some of the most important xenobiotic metabolising enzymes in the cluster are not vital to the organism, and if any of the genes perform critical housekeeping or intermediary metabolism then redundant function must lie in genes beyond this cluster.
- iii) diminishing the problem of individual gene knockdown or knockout in which closely related genes are upregulated in compensation that may mask or dilute phenotypes.

What the approach lacks is the ability to examine the role of individual genes, and this is clearly the next step in the research that would go on to use CRISPR/Cas9 to delete individual genes to reinforce the candidate gene phenotypes identified in the manuscript.

We thank the referee for pointing this out. Indeed it was implicit, and perhaps too obvious to us to mention it specifically, that deleting the whole cluster would give a global view of the in vivo role of the genes. In a sense, it is both a proof of principle, and a first pass approach. Although we functionally expressed all the genes individually in vitro, we can also knock them out individually for a finer analysis. We now mention this future step (insertion line 198 of initial review pdf).

The paper's other novelty has been to challenge some previous conclusions identified through phenotypes produced by RNAi of individual genes, that were not supported by later enzyme activity data. The data also indicates that upregulation of specific P450s in response to xenobiotic exposure is not a guarantee that that P450

is involved in metabolism of the particular xenobiotic and may reflect co-regulation of genes within the cluster.

Before publication, I would like to see some modifications to the manuscript that would make the presentation a little clearer to those not intimate with the field. The level of detail provided in the manuscript is generally sufficient to reproduce the work, particularly if, as suggested below, that the indicated statistical methods were clarified.

These include:

1) I don't understand the prominence of the MS figures Fig4, 5 and 6. They are barely mentioned in the results and discussion and seem to be more suited to the supplementary material. I would thus suggest that they were moved to supplementary or if there is further MS information that this space was used to identify the different xenobiotic metabolites to demonstrate compound detoxification.

We followed common practice by measuring substrate depletion and provided the spectra as evidence that this was done correctly. Figures 4, 5, and 6 have been moved to supplementary material and we corrected the legend to Figure 6 (now Suppl. Fig. 3), with our apologies. Of course we would have preferred to chemically identify all the metabolites, but in the absence of reference metabolites, this would be a major effort of analytical chemistry that is beyond the aim of the present paper.

2) Fig2 may be simplified to remove the reference strain values that by definition are equal to 1 in each case.

Figure 2 and its legend have been simplified as suggested.

3) Supple table 1 should also present the raw data from the larval toxicity assays rather than just the summary data. Include in the table the precise statistical method used to produce the 95% fiducial limits for transparency and reproducibility by others.

The statistical method was detailed in lines 392-396 of the initial review pdf. We presented the bioassay data in Table S1 following common conventions, and the fiducial limits were calculated in the widely used Polo plus program, thus providing all necessary information for others to repeat and verify our experiments. We also now provide the number of concentrations used to establish the log probit lines.

4) An indication as to whether single gene knockouts will be performed or are necessary to validate results further.

We have now indicated that single knockouts are a future aim (insertion line 198 of initial review pdf). We hesitate to say that such work would be a NECESSARY validation, because we feel that it would be difficult to imagine a scenario where the phenotype of the cluster knockout would be contradicted by the phenotype of an

individual member of the cluster, especially because our results are backed up by functional expression of each of the P450s.

Minor modifications:

Line 39 replace “responsible for detoxification” with capable of metabolism

Corrected as suggested.

135 replace detoxification of with tolerance to – the bioassays cannot show a detoxification route specifically

Ditto line 140 and 144 line

Corrected as suggested.

150 include P.polyxenes in front of CYP6B1

Corrected as suggested.

157 remove subjective values such as high – eg the value for cyp6B1 is the highest observed in the samples tested.

Corrected as suggested.

line 160 put in the value of the next highest one to AE19 such that, only CYP6AE19 was shown to metabolise xanthotoxin at an activity >0.? Pmol/

Corrected. We have added the limit of detection in this sentence as well as in the legends of Fig. 3 and Suppl. Table S3

Line 162-165 These activity values indicate very slow turnover even for P450s. How significant are these values compared to the values given by asterix (which I presume are not zero?) Please indicate the cut off where activity is deemed to be negligible.

Indeed, some activities are low, and this is often the case in the P450 literature. Optimization of the multiplicity of infection (MOI) of P450 and reductase, or inclusion of cytochrome b5, might in some cases increase the activity observed in vitro. However, the fact that we can measure P450 spectra and thus provide activities per pmol P450 as well as the stringent controls and significance of the activities over the controls give us great confidence that what we measure is real. We also now provide the very stringent limits of detection for substrate depletion of each compound, calculated as three times the root mean square error of the y residuals of the calibration lines, divided by the slope.

Line 190 Is there any evidence that viability of the knockout would be affected if fed certain foodstuffs. Possibly need to qualify sentence to under these experimental conditions.

This is a very valid comment and we fully agree. We now state more specifically (insert lines 124 of the submission pdf) that viability was normal under our laboratory conditions. While our figure 2 shows that viability is affected by plant allelochemicals, the logical implication that certain host plants would be less well tolerated by the KO larvae is well worth a specific mention. We now do so (insert line 191 of the submission pdf). As a highly polyphagous insect, H. armigera can deal with a wide variety of plant chemistries, and we are going to measure comparative fitness of the cluster KO strain on different host plants. We expect this to provide more clues to the function of the CYP6AE genes, and help refine experiments on individual knock outs.

Line 197 I think the following is correct; the reverse genetic approach has provided in vivo data to confirm involvement of the cluster in xenobiotic tolerance in H. armigera

We agree. The text has been modified as suggested.

Lines 212 215 I would argue that the strategy is not a precise way it provides a global view. The precise way would be to also knock out individual genes to give “full characterisation”. The global strategy is very useful as a first pass, along with transcriptomic and proteomic data.

This was our poor construction of the sentence. The precise way was in reference to the method not its result, which is indeed global. This has now been corrected.

Fig 7 indicate what the scale is on the right

The legend of the present Fig.4 has been revised to indicate the scale: Fold difference vs control (i.e. log₂ base fold changes)..

Reviewer #2 (Remarks to the Author):

The authors used genome editing to knock out a cluster of nine cytochrome P450s (85kb) to demonstrate their cluster contribution to xenobiotic (phytotoxins and pesticides) detoxification, in a major agricultural pest world wide, *Helicoverpa armigera*. Validation was done by parallel in vitro functional studies (expression of recombinant enzymes and characterization of their catalytic properties against same xenobiotics used in bioassays), as well as induction and qPCR experiments. Among claims, an interesting verification that the CYP6AE14 is not involved in gossypol defense (article previously published in Nature, possible RNAi leaking/off target effects). Another interesting claim, the inducibility alone can non be used as evidence for a role in detoxification.

The findings are certainly novel and both outcome and methodology will be of substantial interest to others in the community and the wider field. The paper is going to really influence thinking in the field, and beyond (for example P450s involved in resistance of major vector borne diseases) as several classical approaches, linking gene to phenotype, will have to be re-considered.

The study is technically well executed and of very high quality standards and the manuscript is very well written and clearly presented. The conclusions are well supported by the experimental data and no further experiments are required. I would strongly recommend the acceptance of this manuscript in Nature Communication.

One minor thing would only recommend to the authors to include in their discussion, why they think there is still significant difference in the pesticide resistance levels observed in field populations in association studies (for example indoxacarb resistance and P450s), with the ones obtained after the KO (for example 3.1 – fold for indoxacarb).

*We feel that several factors may lead to such differences: In the KO experiments, we compare susceptibility levels between a reference laboratory strain and the same without nine P450 genes. In the field studies, a reference strain is compared with a survivor strain (i.e. selected) of a genetically distinct (wild) population. By the time indoxacarb resistance is noted in the field, selection has operated on not just a single P450 gene, but on all kinds of other genes as well. Moreover, and perhaps most importantly, P450s other than the CYP6AE may also have been selected by indoxacarb (after all, there are 114 P450 genes in *H. armigera*). To prevent the mistaken conclusion that ONLY CYP6AE17 and 18 would be responsible for indoxacarb resistance, we have followed the referees' advice and added a note of caution (insert lines 231 of the submission pdf).*

Reviewer #3 (Remarks to the Author):

The article describes the use of CRISPR to delete a large cluster of P450 genes, that probably originally arose via duplication, with a view to investigating the role of members of this cluster in the detoxification of various insecticides and phytotoxins. The intro to the article and the description of the experimental flow are largely a pleasure to read for their flow and their clarity, so compliments to the authors for that. Having said that, functional confirmation of the role of P450s in such processes is not new, and neither is the use of CRISPR with two sgRNAs to perform large genomic deletions. Therefore I would have some significant concerns over the novelty of the findings.

*While we are reluctant to make bold claims of priority in the paper, to the best of our knowledge, no cluster of P450 genes has yet been knocked out in non model organisms. In the mouse, KO of three clustered CYP2B genes confirmed their known function, and in the rat, a technical paper showed replacement of five clustered CYP2D genes by the human CYP2D6 but was not followed up by any biological evaluation. We wish to revise the paper by the citation of Itokawa et al (2016) who knocked out CYP9M10 in a pyrethroid-resistant strain of *Culex*, and thus confirmed the known role of this P450 in resistance.*

What would have made this paper much stronger would be to have performed a functional evaluation of individual P450 genes *IN VIVO* in order to unequivocally tease out the roles of different P450s and to determine whether there has been specialisation of function. Instead, having performed the large deletion, the

authors chose to test different P450-toxin combinations in vitro and extrapolate their findings. The conclusions of these findings then made no use at all of the knockout and thus are subject to all the caveats of in vitro experiments such as controlling for copy number, tissue variegation, position effects at insertion site etc.

We did in fact document specialisation of biochemical function by expressing ten CYP6AE enzymes in vitro with a variety of substrates (see also our previous work, ref 22). While this can be considered standard operating procedure for individual P450 enzymes, we used a panel of ten enzymes (+ CYP6B1) and screened it with multiple substrates. To the best of our knowledge, only one study in insects has ever studied more P450s in a single study (Manjon et al. Current Biology 2018). Indeed, in vitro assays of recombinantly expressed P450 enzymes remain a challenging task. To functionally evaluate individual P450 genes in vivo, we would have to knock out each of the ten genes. We now mention that this would certainly be of interest, but we feel that such an effort is beyond the scope of this paper.

MINOR POINTS

Line 175 "based on Ct value" it is not clear from this statement how the quantification of the qRT experiments done and this is a peculiar choice of phrase.

Lines 174-176 of the review pdf were not stringent and not necessary. So, we deleted this sentence.

Discussion is way too long. Lines 20-204 contain too much technical detail about a hypothetical future application of CRISPR editing.

The list of clusters has been replaced by a shorter statement about their characteristics in the H. armigera genome and a reference to the genome paper.

Lines 216-243 read more like a review article of P450s and are out of place here.

Because of the vast literature on xanthotoxin and Lepidoptera, we felt that it was important to give credit and place our results in the proper perspective, but in accordance with the reviewer's comment, we have distilled this paragraph to the essential.

More might have been made about the frequency with which large deletions could be recovered and what this means for the field in general.

We inserted a rough estimate of the frequency of deletion recovery in line 188 of the initial review pdf. We would hesitate to draw conclusions, either encouragements or warnings to others, because we feel that, as with other such techniques, practice and experience will probably improve success rates. Moreover, the size and development of embryos is different in each lepidopteran species, so that our results would only serve as single case study for H. armigera.

I am happy for my review to be non-anonymised.

Tony Nolan

REVIEWERS' COMMENTS:

Reviewer #1 (Remarks to the Author):

The authors have done an excellent job to address satisfactorially all the concerns that were put forward by the three reviewers.

Reviewer #2 (Remarks to the Author):

I believe the authors have made sufficient amendments and improved the manuscript, in line with the reviewers recommendations. The lack of individual gene evaluation also by CRISPR/Cas9 that has been raised, is a next step in the research, but realistically not applicable for inclusion in this manuscript, given the timelines and amount of work involved, and the authors clearly discuss this point, which is satisfactory I believe. I recommend the acceptance of the manuscript for publication, with no further corrections.

Reviewer #3 (Remarks to the Author):

I am happy that the comments have been sufficiently addressed. I would have preferred it if the limitations of the in vitro analysis had been more explicitly stated but am not going to insist on it.

Responses to the referees

REVIEWERS' COMMENTS:

Reviewer #1 (Remarks to the Author):

The authors have done an excellent job to address satisfactorily all the concerns that were put forward by the three reviewers.

Response: Thanks for positive feedback.

Reviewer #2 (Remarks to the Author):

I believe the authors have made sufficient amendments and improved the manuscript, in line with the reviewers recommendations. The lack of individual gene evaluation also by CRISPR/Cas9 that has been raised, is a next step in the research, but realistically not applicable for inclusion in this manuscript, given the timelines and amount of work involved, and the authors clearly discuss this point, which is satisfactory I believe. I recommend the acceptance of the manuscript for publication, with no further corrections.

Response: Thanks for positive feedback.

Reviewer #3 (Remarks to the Author):

I am happy that the comments have been sufficiently addressed. I would have preferred it if the limitations of the in vitro analysis had been more explicitly stated but am not going to insist on it.

Response: Thanks for positive feedback. We stated this in the results section: "Successful expression of CYP6AE P450s was demonstrated by their reduced CO-difference spectrum and model substrate activity tests (22). While higher enzyme activities might be achieved with different experimental conditions (multiplicity of infection, addition of cytochrome b5), the in vitro assays allow comparisons of the catalytic competence of the individual CYP6AE P450s."